# COVID-19 lockdown emission reductions have the potential to explain over half of the coincident increase in global atmospheric methane

David S. Stevenson[1], Richard G. Derwent[2], Oliver Wild[3], William J. Collins[4]

[1]School of GeoSciences, The University of Edinburgh, Edinburgh EH9 3FF, UK
[2]rdscientific, Newbury, UK
[3]Lancaster Environment Centre, Lancaster University, Lancaster, UK
[4]Department of Meteorology, University of Reading, Reading, UK

*Correspondence to*: David S. Stevenson (David.S.Stevenson@ed.ac.uk)

**Abstract.** Compared to 2019, measurements of the global growth rate of background (marine air) atmospheric methane rose
by 5.3 ppb/yr in 2020, reaching 15.0 ppb/yr. Global atmospheric chemistry models have previously shown that reductions in nitrogen oxide (NOx) emissions reduce levels of the hydroxyl radical (OH), and lengthen the methane lifetime. Acting in the opposite sense, reductions of carbon monoxide (CO) and non-methane volatile organic compound (NMVOC) emissions increase OH and shorten methane's lifetime. Using estimates of NOx, CO and NMVOC emission reductions associated with COVID-19 lockdowns around the world in 2020, together with model-derived regional and aviation sensitivities of methane
to these emissions, we find that NOx emissions reductions led to a 4.8 (3.8 to 5.8) ppb/yr increase in the global methane growth rate. Reductions in CO and NMVOC emissions partly counteracted this, changing (reducing) the methane growth rate by -1.4 (-1.1 to -1.7) ppb/yr (CO) and -0.5 (-0.1 to -0.9) ppb/yr (NMVOC), yielding a net increase of 2.9 (1.7 to 4.0) ppb/yr. Uncertainties refer to ±1 standard deviation model ranges in sensitivities. Whilst changes in anthropogenic emissions related to COVID-19 lockdowns are probably not the only important factor that influenced methane during 2020, these results indicate
that they have had a large impact, and that the net effect of NOx, CO and NMVOC emissions changes can explain over half of the observed 2020 methane changes. Large uncertainties remain in both emissions changes during the lockdowns and methane's response to them; nevertheless, this analysis suggests that further research into how atmospheric composition changed over the lockdown periods will help us to interpret past methane changes and to constrain future methane projections.

## 1 Introduction

Methane is a powerful greenhouse gas and important precursor of tropospheric ozone; both are key air pollutants and short-lived climate forcers (SLCFs). Several factors in addition to rising anthropogenic methane emissions have influenced the evolution of atmospheric methane from its pre-industrial level of ~700 ppb to its present-day value of over 1900 ppb. The Intergovernmental Panel on Climate Change's Sixth Assessment Report (Szopa et al., 2021) assessed how changes in emissions of NOx, CO, and NMVOCs have contributed to historical changes in methane, through their impacts on OH, the
main sink for methane. A range of modelling studies have explored these indirect impacts on methane (e.g., Shindell et al.,

2005, 2009; Stevenson et al. 2013; Thornhill et al., 2021). For example, the Atmospheric Chemistry and Climate Model Intercomparison Project found that 1850-2000 increases in anthropogenic NOx emissions had reduced year 2000 methane levels by 955 ppb, whilst growing emissions of CO and NMVOCs had increased methane by 150 ppb and 59 ppb, respectively (Table 7 of Stevenson et al., 2013). These results have quite large uncertainties (at least ±10%, based on the model range in Stevenson et al. 2013), but indicate that non-methane (especially NOx) emissions have had very significant impacts on methane. Better understanding of what controls methane and its evolution is vital for progress towards the Paris Climate Agreement target that seeks to limit warming to 1.5°C above pre-industrial levels.

Following the onset of the COVID-19 pandemic in early 2020, the trace gas composition of the global atmosphere changed substantially. Atmospheric nitrogen oxide levels reduced as surface and aviation NOx emissions fell (Bauwens et al., 2020; Cooper et al., 2022), whilst the measured growth rate of methane ($CH_4$) rose sharply in 2020 (Laughner et al., 2021). The observed NOx changes are clearly linked to falls in emissions resulting from lockdowns, but the driver of the methane increases is less clear, with some studies discussing causes related to decreases in OH (e.g., Weber et al., 2020; Laughner et al., 2021) while others suggest rises in sources (e.g., Feng et al., 2022). Methane, NOx, CO and NMVOCs are linked via the oxidising capacity of the atmosphere, specifically by the abundance of the hydroxyl (OH) radical. The response of global atmospheric chemistry to the large lockdown perturbation since early 2020 provides an opportunity to explore the sensitivity of the NOx-CO-NMVOC-OH-$CH_4$ system, and compare models and observations. Here we use model-derived sensitivities of global methane to NOx, CO and NMVOC emissions, together with estimated changes in anthropogenic emissions of these species related to the COVID-19 lockdowns, to calculate estimated impacts from lockdown emissions changes on the growth rate of global methane, and compare this to observations.

## 2 Measurements of atmospheric methane and nitrogen oxides

Recent methane measurements from the US National Oceanographic and Atmospheric Administration (NOAA) show that the atmospheric (marine air background) methane growth rate rose sharply from 9.7 ppb/yr in 2019 to 15.0 ppb/yr in 2020, higher than any preceding annual value in the NOAA record, that started in 1984 (Dlugokencky, 2022). Many of the earlier large year-to-year jumps in methane's growth rate relate in part to variability in climate and emissions associated with El Niño Southern Oscillation (ENSO), and in part because of modulation of methane's main sink, oxidation by OH (Turner et al., 2018; Zhao et al., 2020). The start of 2020 marked the onset of a La Niña that has persisted into 2022. Past La Nina's have not always shown clear links with methane's growth rate, and the influence of the current ENSO phase on methane is uncertain.

Measurements of nitrogen dioxide ($NO_2$) from satellite instruments and nitrogen monoxide (NO) and $NO_2$ from surface sites show that levels of atmospheric NOx (NO + $NO_2$) dramatically fell globally during 2020 (Bauwens et al., 2020; Laughner et al., 2021; Cooper et al., 2022). This was driven by COVID-19 lockdowns around the world that reduced emissions, mainly from transportation (Venter et al., 2020; Lamboll et al., 2021; Doumbia et al., 2021).

## 3 Sensitivity of global methane to NOx, CO and NMVOC emissions

Global atmospheric chemistry model simulations indicate that decreases in NOx emissions lead to reductions in OH and increases in the global methane lifetime (Prather, 1994; Derwent et al., 2001; Wild et al., 2001; Stevenson et al., 2004; Weber et al., 2020). Similarly, decreases in CO and NMVOC emissions lead to increases in OH and decreases in methane lifetime (Derwent et al., 2001; Wild et al., 2001). Although methane has an atmospheric lifetime of about 10 years, models show that its peak response occurs within a few months of the cessation of a sudden short-lived (month- or year-long) pulse of extra emissions (Derwent et al., 2001; Wild et al., 2001; Stevenson et al., 2004). This indicates that the impacts on methane from the sudden changes in emissions associated with lockdowns will have had rapid impacts on methane's growth rate.

We first illustrate the basis of our approach by describing the model experiments performed by Derwent et al. (2001), who conducted a series of simulations with the global tropospheric chemistry model STOCHEM to quantify the impact of NOx emissions on methane. They compared a 4-year long base simulation with a perturbation simulation that was identical apart from an enhancement in NOx emissions of magnitude 1 $Tg(NO_2)$, added during the first month with the Northern Hemisphere surface anthropogenic NOx emissions distribution. The extra NOx produced a short-lived increase in OH, and this led to a rapid depletion of global methane, which peaked with a magnitude of around 0.39 $Tg(CH_4)$ after about six months. The methane deficit then exponentially decayed with an e-folding timescale of about 12 years (the methane perturbation lifetime, $\tau$), with methane levels returning towards their base values. Wild et al. (2001) conducted similar experiments, with year-long emissions perturbations using a different model (UCI CTM), and found very similar behaviour but with slightly larger sensitivities: 1 $Tg(NO_2)$ from global fossil fuel sources yielded a 0.55 Tg depletion of $CH_4$. These studies also investigated the impact of CO and NMVOC emissions. Changes in global methane burden (Tg) are converted to changes in tropospheric mole fraction (ppb) using the total atmosphere mass of 5.113 x $10^9$ Tg and a fill factor of 0.973 for conversion of a total atmosphere abundance to a tropospheric abundance (Prather et al., 2012). We assume the troposphere is well mixed, so surface changes will be the same as whole troposphere changes.

More recently, Fry et al. (2012) analysed results from 11 global models that took part in the Hemispheric Transport of Air Pollutants (HTAP) study in order to isolate the impacts on methane of anthropogenic NOx, CO and NMVOC emissions from Europe (EU), North America (NA), South Asia (SA) and East Asia (EA). We utilise that ensemble of model results here; model descriptions are given in Fiore et al. (2009). Models performed a base simulation, and a series of further repeat simulations with 20% lower anthropogenic emissions for each species for each region. In addition to the 20% regional emission reduction experiments, some models also performed global 20% emission reduction experiments (Wild et al., 2012). The results from Fry et al. (2012) and Wild et al. (2012) and the details of our analysis are presented in the Supplementary Material. Four models include results from all the regional and global perturbation simulations: FRSGCUCI-v01, GISS-PUCCINI-modelE, MOZARTGFDL-v2, and TM5-JRC-cy2-ipcc-v1. We calculate a 'four model mean' (4MM) based on these model results. We also show results from the other models to illustrate the range of model behaviour, and show 'multi-model mean' (MMM) results from all available simulations.

In the HTAP simulations, methane was fixed as a prescribed boundary condition, precluding direct diagnosis of changes in methane. However, methane changes can be diagnosed indirectly, by analysing the methane lifetime associated with the tropospheric OH sink in each run. We convert these to whole atmosphere lifetimes by assuming fixed lifetimes for methane losses to soils (150 yr), reaction with chlorine radicals (200 yr) and in the stratosphere (120 yr) (Prather et al., 2012). The HTAP experiments also included a global methane perturbation simulation – allowing the methane feedback factor and perturbation lifetime to be calculated (Prather, 1994; Holmes, 2018). Figure 1 shows whole atmosphere and perturbation methane lifetimes for the HTAP models, with MMM values of 8.3 years and 10.9 years, respectively.

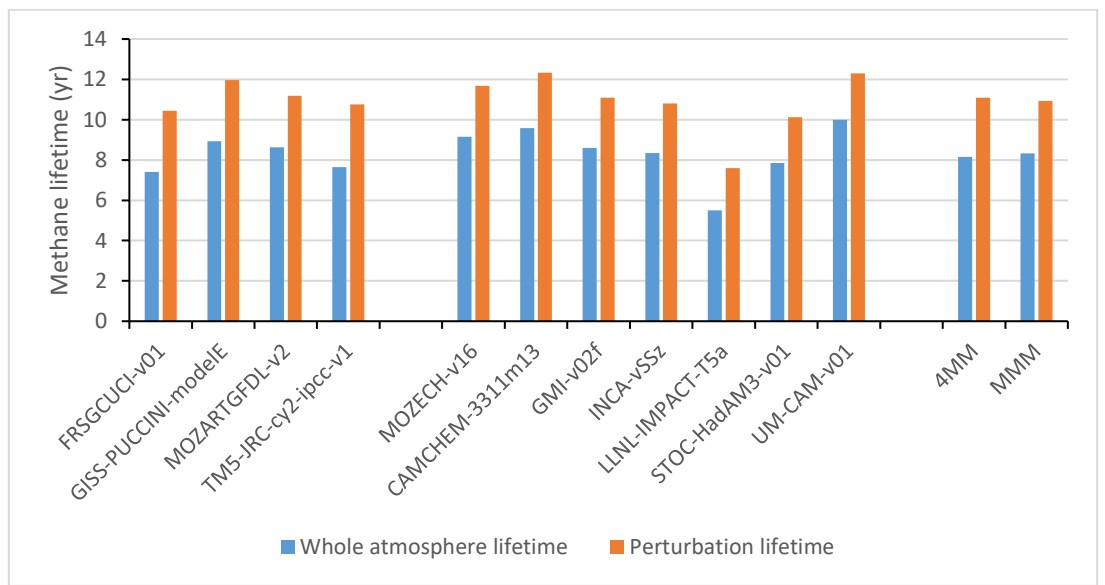

**Figure 1.** Whole atmosphere and perturbation methane lifetimes (years) for the HTAP models, together with the four model mean (4MM) of the core models (four models on the left), and the multi-model mean (MMM). Perturbation lifetime derived from 20% methane reduction experiments (see Supplementary Table S4).

Differences between simulations yielded the change in methane lifetimes due to changes in regional/global emissions. From these changes in methane lifetime, the equilibrium change in methane was calculated; that is the change in methane that would have occurred if methane levels had been free to respond (e.g., see Stevenson et al., 2013). In model simulations where methane is not prescribed, methane adjusts towards equilibrium with an e-folding timescale given by its perturbation lifetime (Derwent et al., 2001; Wild et al., 2001; Holmes, 2018). We convert equilibrium methane changes derived from sustained changes in emissions to the equivalent methane response for a pulse of emissions for each experiment. We use each model's perturbation lifetime to calculate the fraction of the equilibrium response that would have been reached after one year; e.g., for the multi-model mean (MMM) methane perturbation lifetime of 10.9 years (Figure 1) this fraction is $(1-e^{-1/\tau}) = 8.8\%$. This method is appropriate because we compare to changes in the observed annual growth rate, and is justified by the rapid response of global

methane seen in transient model simulations where methane is free to respond, and because the largest lockdown emissions' perturbations occurred in the first half of 2020. We normalise results to produce global methane sensitivities per Tg of gas emitted for each HTAP region and globally for each model. Figures 2, 3 and 4 show global methane sensitivities for NOx, CO and NMVOC emissions, respectively.

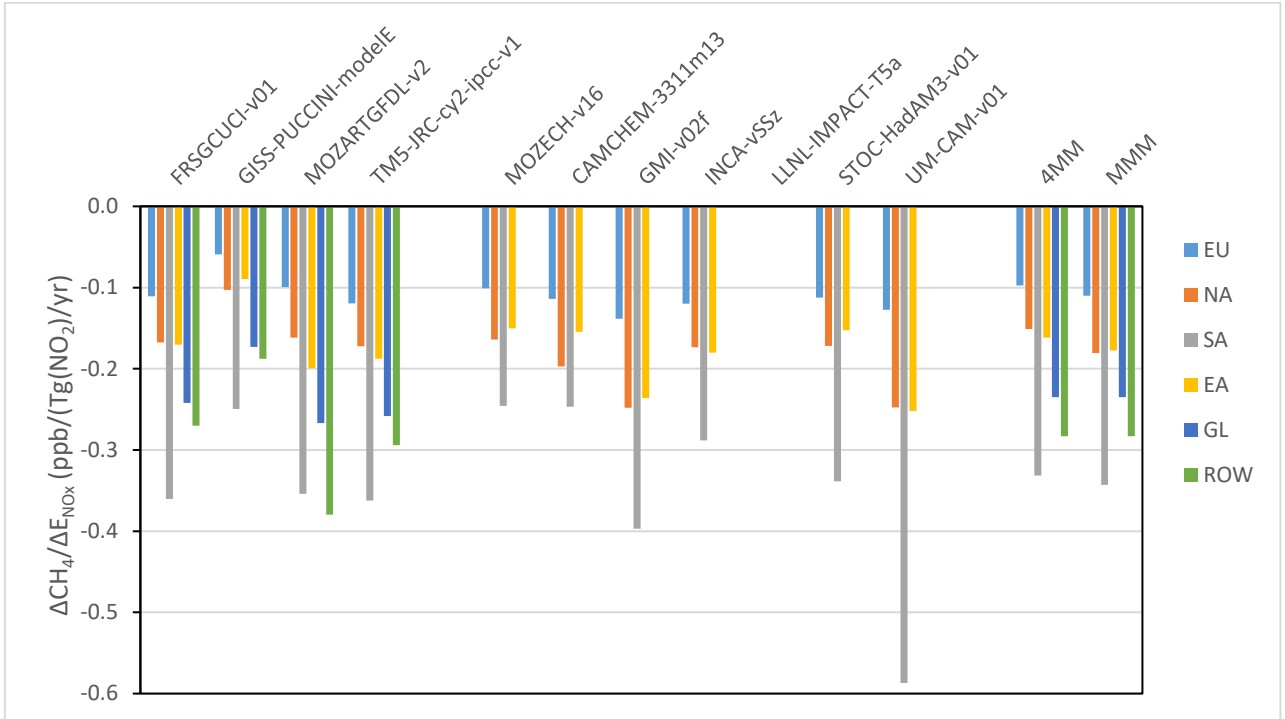

**Figure 2.** Sensitivity of global methane (ppb) to changes in anthropogenic NOx emissions (Tg(NO$_2$)/yr), derived from 20% reduction experiments performed by the HTAP models for four regions (Europe, EU; North America, NA; South Asia, SA; and East Asia EA), and also globally (GL), and for the ROW (everywhere outside the four HTAP regions). Global (and hence ROW) results are only available for the four core models, shown on the left of the figure. Also shown are the 4MM and MMM. There are no results for the LLNL-IMPACT-T5a model for NOx; it is included to maintain consistency with Figures 3 and 4.

Figure 2 shows relatively consistent responses to NOx emissions, with all models least sensitive to EU NOx emissions and most sensitive to SA, with NA and EA in between. Global and ROW sensitivities are relatively high. The 4MM sensitivities are slightly lower than the MMM.

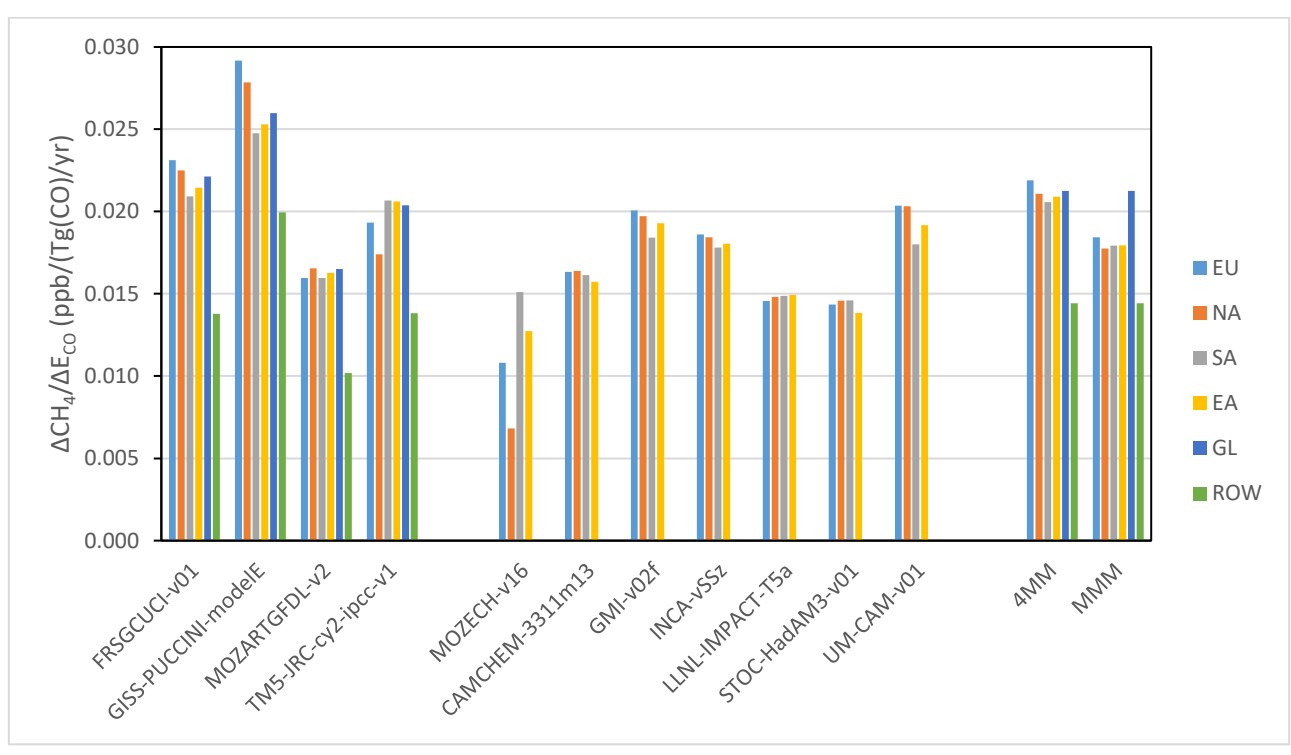

140        **Figure 3.** As Figure 2, but methane sensitivities for changes in surface anthropogenic CO emissions (Tg(CO)/yr).

Figure 3 shows relatively consistent behaviour across the models for CO, with less variation between regions, reflecting the longer lifetime of CO, which makes the location of emissions less important. The ROW sensitivities, inferred from the global results are relatively low. The 4MM sensitivities for CO are slightly larger than the MMM values.

145

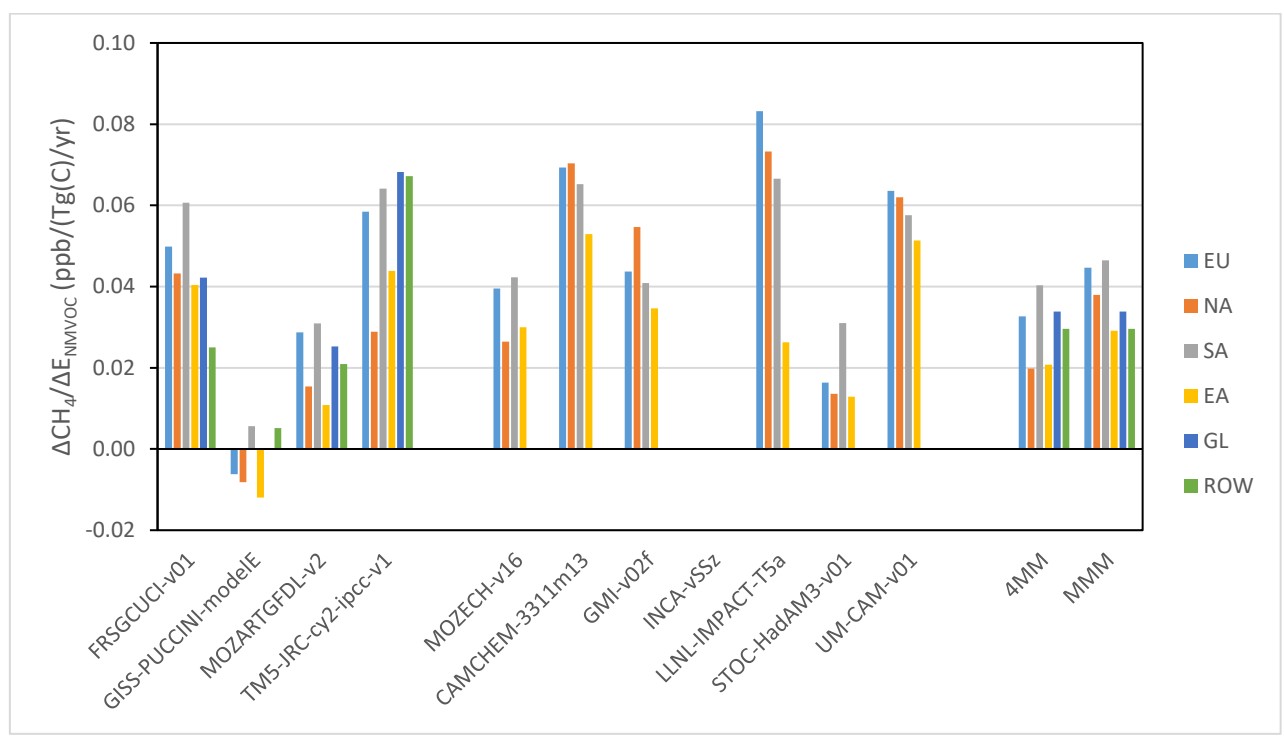

**Figure 4.** As Figure 2, but methane sensitivities for changes in surface anthropogenic NMVOC emissions (Tg(C)/yr). There are no results for the INCA-vSSz model for NMVOCs.

Figure 4 shows more divergence in model response to NMVOC emissions, with one model (GISS-PUCCINI-modelE) displaying an opposite sensitivity to the other models (apart from for SA and ROW emissions), and some models showing quite large sensitivities, whilst others are small. This probably reflects differing methods of representing NMVOCs in each model, in terms of both the number of species, grouping together of species, and the sophistication of their oxidation chemistry. Somewhat fortuitously, the 4MM and MMM are similar.

The HTAP experiments used 2001 as their base year, prescribing global methane to be 1760 ppb, and each model used their own best estimates of global 2001 emissions. In 2020, surface level background global mean methane was ~1870 ppb, and emissions of NOx, CO and NMVOCs had changed relative to 2001. Sensitivities of methane to emissions derived from the HTAP results will differ somewhat from those that would be found if 2020 conditions were used, and this represents an important caveat to our results. However, these differences are unlikely to be substantial, and no more up-to-date multi-model study of the impacts of regional NOx, CO and NMVOC emissions on methane has been published to date, so it represents our best source of information in the literature.

The HTAP perturbation experiments were for all anthropogenic emissions, including aviation, which is a significant NOx source. Similar model simulations have calculated the sensitivity of methane to aviation NOx emissions, and this allows us to separate out the effects from aviation. Wild et al. (2001) and Stevenson et al. (2004) conducted pulse experiments adding NOx

using the global aviation NOx emission distribution, and found a peak impact on global methane of about 2.5-2.6 Tg (equivalent to mole fractions of 0.88-0.92 ppb) for a 1 $Tg(NO_2)$ emission perturbation. Stevenson and Derwent (2009) also found spatial variation in sensitivity for aviation NOx, with the more sensitive regions tending to have lower background NOx levels. The most up-to-date study of aviation NOx is Lee et al. (2021), who assessed multi-model results using sustained emissions changes, similarly to the HTAP study. Lee et al. (2021) report (their Table 3) a methane radiative forcing sensitivity to aviation NOx emissions of -15.8 mW m$^{-2}$ $(Tg(N)$ yr$^{-1})^{-1}$. We convert this to a methane mole fraction sensitivity to NOx emissions using the relationship between changes in mole fraction and radiative forcing given by Myhre et al. (1998), and then, using a similar methodology to that described above, to the equivalent response for a pulse of emissions. This yields a sensitivity of methane to a pulse change in aviation NOx emissions of 1.12 ppb $(CH_4)/Tg(NO_2)$ yr$^{-1}$, similar to, but slightly higher than results from the earlier studies. Lee et al. (2021) also report a 95% likelihood range on the radiative forcing sensitivity, which translates to a standard deviation of 0.21 ppb $(CH_4)/Tg(NO_2)$ yr$^{-1}$, which we take to be a representative uncertainty for the mole fraction sensitivity to aviation NOx emissions.

## 4 COVID-19 lockdown impacts on emissions

Lamboll et al. (2021) compiled estimates of the impact of COVID-19 lockdowns on global anthropogenic NOx, CO and NMVOC emissions, as monthly mean time series, with spatial resolution 0.5° latitude by 0.5° longitude. We use these data to calculate the difference in surface and aviation NOx emissions between 2019 (pre-lockdown) and 2020 for the four HTAP regions, as well as globally, and hence for the 'Rest of the World' (ROW) region (i.e. everywhere beyond the four HTAP regions). The annual reduction in global surface NOx emissions from 2019 to 2020 was about 19.38 $Tg(NO_2)$, or 15%. Lamboll et al. (2021) also compiled data on aviation emissions, estimating a global reduction of about 0.83 $Tg(NO_2)$, or 23%. Global and regional annual changes in NOx, CO and NMVOC emissions are summarised in Table 1.

| | Surface NOx | | Aviation NOx | | Surface CO | | Surface NMVOC | |
|---|---|---|---|---|---|---|---|---|
| | $Tg(NO_2)$ | *%* | $Tg(NO_2)$ | *%* | $Tg(CO)$ | *%* | $Tg(C)$ | *%* |
| Global emissions (GL) | -19.38 | *-14.6* | -0.83 | *-23.2* | -73.38 | *-12.9* | -15.65 | *-9.9* |
| Europe (EU) | -2.65 | *-15.8* | -0.23 | *-23.4* | -6.09 | *-18.1* | -1.71 | *-10.7* |
| North America (NA) | -2.55 | *-18.7* | -0.23 | *-23.1* | -7.49 | *-14.0* | -1.56 | *-12.3* |
| South Asia (SA) | -3.78 | *-16.5* | -0.02 | *-23.0* | -16.76 | *-16.1* | -4.34 | *-16.1* |
| East Asia (EA) | -4.40 | *-11.8* | -0.08 | *-22.9* | -24.58 | *-12.5* | -2.41 | *-6.5* |
| Rest of the World (ROW) | -6.00 | *-14.3* | -0.28 | *-23.1* | -18.46 | *-18.5* | -5.63 | *-8.6* |


**Table 1.** Changes in global and regional annual anthropogenic emissions from 2019 to 2020 (in Tg and as a percentage of 2019), assumed to be associated with COVID-19 lockdowns. The Rest of the World (ROW) is defined as everywhere apart from the four HTAP regions. Derived from data in Lamboll et al. (2021).

**5 Impacts of reduced lockdown emissions on global methane**

To calculate an approximate impact of the lockdown emission reductions on global methane, we simply multiply the regional/aviation sensitivities and emissions changes and sum over the globe. To calculate ROW contributions, we assume that the global sensitivity values can be linearly constructed from the four regions and the ROW, weighting each region by its emissions.

An additional complication is that the regional sensitivities from HTAP for anthropogenic NOx emissions shown in Figure 2 are for a combination of surface and aviation sources. The percentage changes in emissions related to lockdowns for surface and aviation NOx emissions differ (Table 1). We first calculate contributions to the global change in methane using the regional sensitivities derived from HTAP (Figure 2) to account for the changes in regional surface emissions. For example, for Europe, surface NOx emissions reduced by 2.65 $Tg(NO_2)$, or 15.8%. By using this 15.8% reduction with the EU sensitivity, we account

for an annual 15.8% reduction in aviation NOx emissions (0.15 $Tg(NO_2)$) from the region. However, the total change in aviation NOx emissions over Europe is 23.4%, or 0.23 $Tg(NO_2)$, so another 0.08 $Tg(NO_2)$ needs to be included. Globally, an extra 0.31 $Tg(NO_2)$ needs adding (Supplementary Table S11b), above that already accounted for by using the HTAP regional sensitivities. We use the global sensitivity to aviation NOx emissions derived from Lee et al. (2021) (see Section 3: 1.12 ± 0.21 ppb $(CH_4)/Tg(NO_2)$ yr$^{-1}$) with this value to derive an extra aviation component of 0.34 ± 0.06 ppb $(CH_4)$ (Figure 5 and

Table 2).

Figure 5 shows calculated contributions to the global methane growth rate from changes in NOx emissions for each of the HTAP models, together with the 4MM and MMM values. Equivalent results for CO and NMVOCs are shown in Figures 6 and 7, respectively. Table 2 summarises the regional and aviation components for all emissions, using results from the 4MM. We find that reduced NOx emissions during lockdown increased the methane growth rate in total by 4.8 ± 1.0 ppb/yr (4MM; a slightly larger impact of 5.0 ± 1.0 ppb/yr is found for the MMM). South Asia is the largest contributing HTAP region, although this is exceeded by the impact from NOx emissions changes from outside the four HTAP regions. Aviation NOx is also an important contributor, making up about 1/5 of the total from NOx. Reduced CO emissions partly counteracted this positive impact on the methane growth rate, with an overall impact of -1.4 ± 0.3 ppb/yr (4MM; a slightly smaller impact of -1.3 ± 0.3 ppb/yr is found for the MMM). East Asia, followed by South Asia, are the largest contributing regions. Reduced NMVOC emissions had an additional effect in the same sense as CO, but about one third smaller, and with a larger uncertainty. The overall impact from NMVOC was -0.5 ± 0.4 ppb/yr (4MM; slightly larger value for MMM: -0.6 ± 0.4 ppb/yr).

We find a net total impact on methane of 2.9 ± 1.1 ppb/yr (4MM; 3.2 ± 1.1 ppb/yr MMM), with the largest contributing region overall being ROW, followed by South Asia. Aviation NOx changes make up about 30% of this net total.



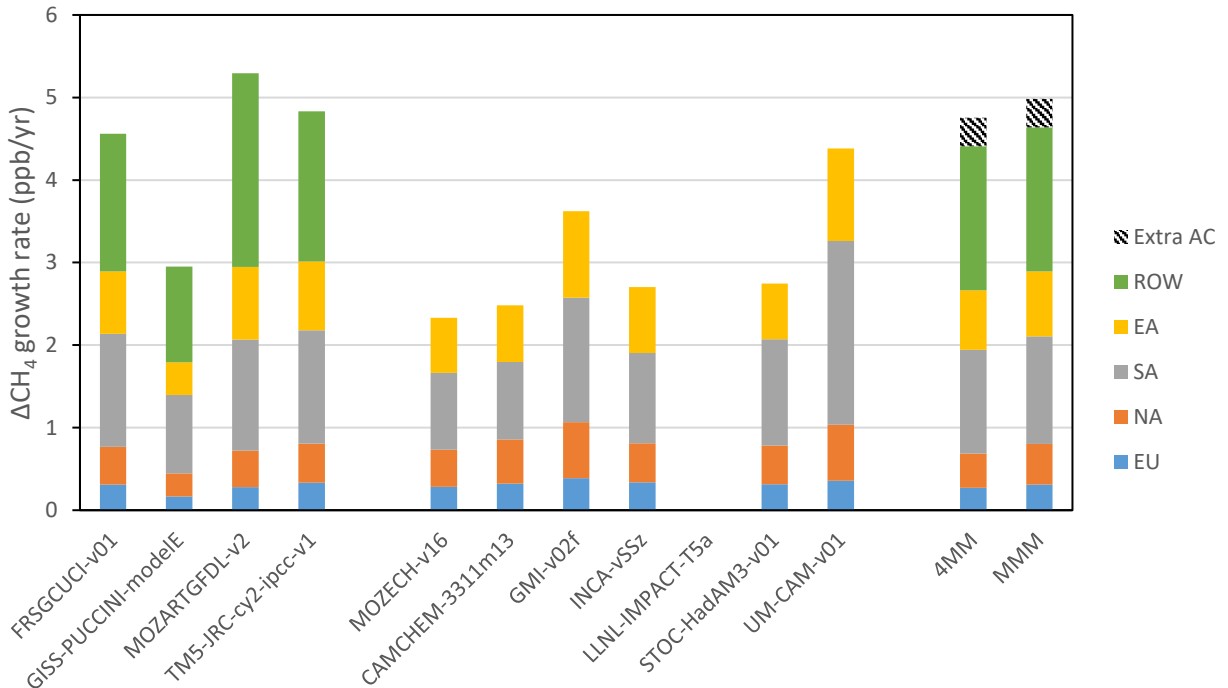

**Figure 5.** Calculated changes in global methane growth rate from changes in anthropogenic NOx emissions during the 2020 lockdown, for each of the HTAP models. Also shown are values for the mean of the four core models (shown on left) (4MM) that reported results for all simulations, together with multi-model mean (MMM) results based on all available models. The four core models included global experiments, allowing calculation of ROW contributions. Regional contributions partially include aviation NOx; a further contribution from aviation (Extra AC) is also shown for the multi-model results (see text for details).

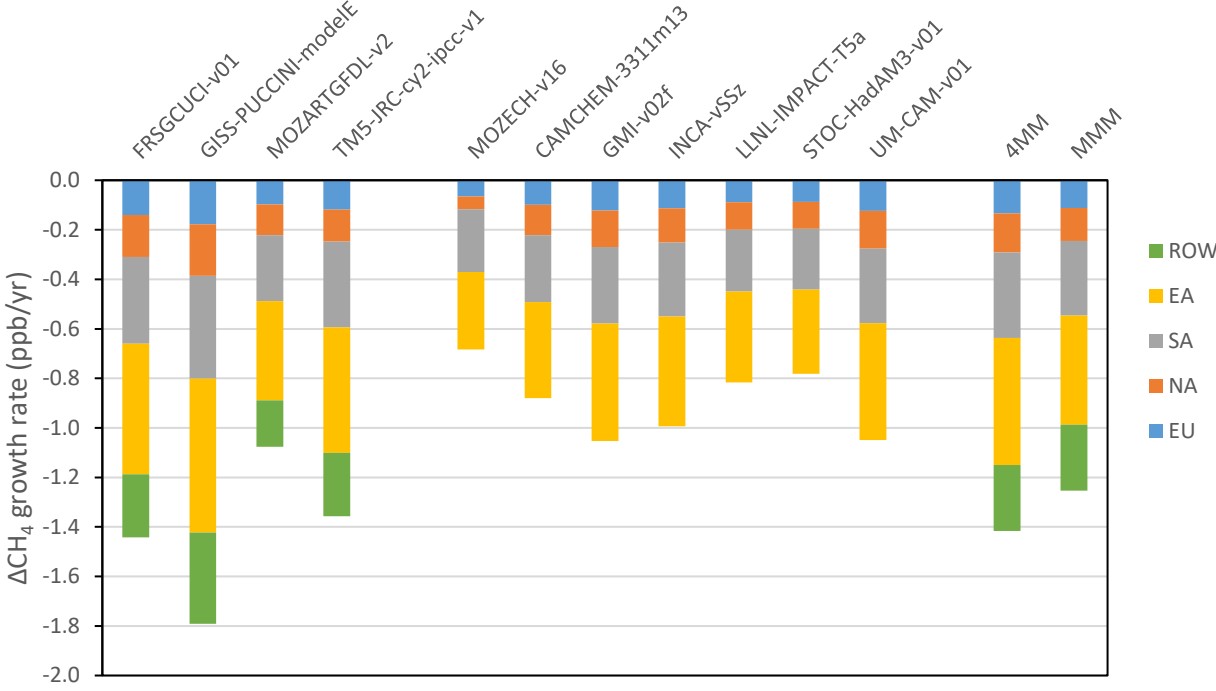


**Figure 6.** As Figure 5, but for CO emissions.

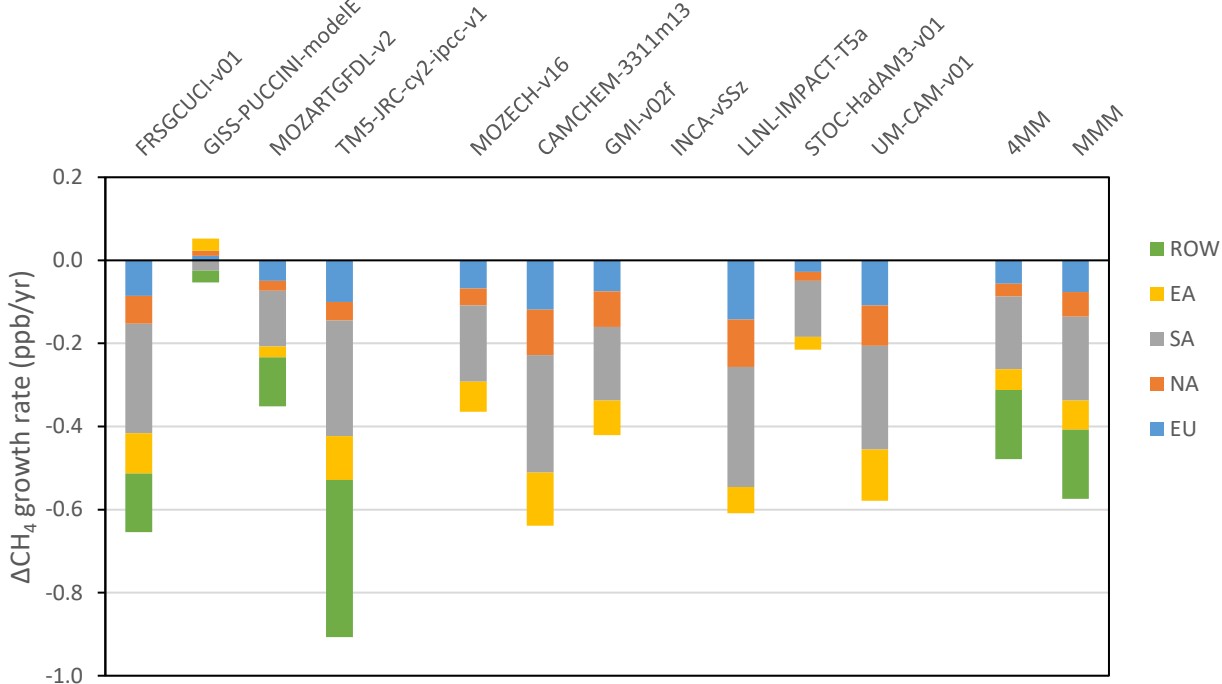

255          **Figure 7.** As Figure 5, but for NMVOC emissions.

|  | **NOx** | | | **CO** | | | **NMVOC** | | | **Total** | | |
|---|---|---|---|---|---|---|---|---|---|---|---|---|
| **Europe** | 0.27 | ± | 0.07 | -0.13 | ± | 0.03 | -0.06 | ± | 0.05 | 0.08 | ± | 0.10 |
| **N America** | 0.41 | ± | 0.09 | -0.16 | ± | 0.04 | -0.03 | ± | 0.03 | 0.22 | ± | 0.10 |
| **S Asia** | 1.26 | ± | 0.21 | -0.34 | ± | 0.06 | -0.17 | ± | 0.12 | 0.74 | ± | 0.25 |
| **E Asia** | 0.72 | ± | 0.22 | -0.51 | ± | 0.09 | -0.05 | ± | 0.06 | 0.15 | ± | 0.25 |
| **ROW** | 1.75 | ± | 0.49 | -0.27 | ± | 0.07 | -0.17 | ± | 0.15 | 1.31 | ± | 0.51 |
| **Aviation (extra)** | 0.34 | ± | 0.06 | | | | | | | 0.34 | ± | 0.06 |
| **Aviation (total)** | (0.93 | ± | 0.18) | | | | | | | (0.93 | ± | 0.18) |
| **Total** | 4.75 | ± | 1.02 | -1.42 | ± | 0.29 | -0.48 | ± | 0.39 | 2.86 | ± | 1.13 |

**Table 2.** Summary of impacts on the 2020 global methane growth rate (ppb/yr) relative to 2019 due to COVID-19 lockdown
emission reductions based on 4MM results. Values for total aviation NOx are bracketed as they are already partly included in
the regional values; the additional aviation component not included in the regional values (Aviation (extra)) is also shown.

## 6 Discussion and Conclusions

These model-derived results can be compared to the observed increase in methane growth rate from 2019 to 2020 of 5.3 ppb/yr, and suggest that lockdown emission changes in NOx, CO and NMVOCs can explain 54 ± 21 % (4MM; MMM: 60 ± 21 %) of this increase. Uncertainties are standard deviations of the HTAP (Fry et al., 2012) and aviation NOx (Lee et al. 2021) model's sensitivity ranges. No uncertainty estimate is included here for the magnitude of lockdown emissions changes, which is probably similar in magnitude. Our results have several important caveats, and refinements to this relatively simply derived estimate will need to account for a number of additional complications. The emission changes have temporal structure (Lamboll et al., 2021), as do the sensitivities of methane to NOx, CO and NMVOCs, and these will interact. One study has reported a reduction in lightning during 2020 (Vasquez, 2022), which may contribute much like reductions in aircraft NOx. The regional sensitivities derived here are based on emissions changes with the spatial distributions and base magnitudes of the 2001 anthropogenic emissions, rather than a 2020 emissions baseline and the actual changes during lockdown. Given non-linearities in the response of OH to emissions, the real sensitivities are likely to be slightly different to those calculated here, and this increases the uncertainty in our results. Detailed modelling of the lockdown period is starting to explore these effects (Weber et al., 2020; Miyazaki et al., 2021). There is also spatio-temporal structure in the observed methane changes (e.g., Laughner et al., 2021; Feng et al., 2022) that will yield further information. There are undoubtedly several other factors, in addition to changes in anthropogenic NOx, CO and NMVOC emissions that influenced methane during 2020. Nevertheless, it seems likely that the dramatic reductions in these emissions, especially NOx, brought about by the COVID-19 lockdowns can explain a large component of the surge in methane growth rate seen during 2020. These influences on methane related to changes in OH need to be carefully accounted for in any attribution study that attempts to explain the recent observed dramatic changes in methane.

## Author contributions

DSS wrote the text and performed the main analysis. OW and WJC performed additional analysis and commented on the text. RGD commented on the text.

## Competing interests

The authors declare that they have no conflict of interest.

**Code/data availability**

Original data used here are all freely available in the cited references. All calculations are detailed in the Supplementary
Material.

**Acknowledgements**

This work was partly supported by the Natural Environment Research Council (NE/S009019/1) and the Royal Society (IES\R3\193183). We acknowledge all the modellers who contributed results to the HTAP Phase 1 study: without those results, this work would not have been possible. Jize Jiang is thanked for his technical help with the analysis.

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
