# Peer review of "COVID-19 lockdown emission reductions can explain over half of the coincident increase in global atmospheric methane"

_Atmospheric Chemistry and Physics, 2021_

## Author Comment (AC1)

We would like to thank the three referees for their invaluable comments. The main points requiring a response are repeated below in italics, followed by our responses.

**Referee #1**

*First, the authors did not assess the impact of other species whose emissions may also have reduced together with NOx during COVID-19, for example, CO and NMVOC. Reduced emissions of these species have an opposite effect to that of NOx, that is, to increase OH and reduce CH4 lifetime. According to Fig 3 of Lamboll et al. (2021), emissions of CO and NMVOC were also substantially reduced though with slightly smaller fractions compared to NOx. Fry et al. (2012) showed with their model that the net effect on OH is close to 0 with compensating effects from a combined 20% reduction of NOx, CO, and NMVOC. So, the author's argument can be much stronger if they can estimate the net chemical effect of the COVID-19 emission perturbation.*

**Response**

This is a good point. We have now applied essentially the same methodology to assess the impacts of CO and NMVOC emissions reductions associated with lockdowns. We use sensitivities calculated by Fry et al. (2012) for the four regions considered within their HTAP study, together with regional and global estimates of lockdown emissions reductions from Lamboll et al. (2021). For emissions changes outside the four regions, we use a global sensitivity value. Results for CO are shown in Table 1 and NMVOC in Table 2. This method only uses sensitivities from Fry et al. (2012), whereas in our original submission we used sensitivities for NOx from some earlier studies as well. For the sake of consistency, we also show in Table 3 revised values for surface NOx using the same methodology as for CO and NMVOC.

In addition, we now include an uncertainty range on our values, based on the spread (±1 Standard Deviation) of sensitivities found within the 11 HTAP models. This is not a full assessment of uncertainty as we don't consider uncertainty in the lockdown emissions reductions.

We find that the impact on global methane from surface NOx emission reductions during lockdown is +3.5 ppb, with a range of +2.4 to +4.5 ppb (Table 3). Adding the component from aircraft NOx (+0.7 ppb) brings the total NOx effect to +4.2 ppb (range +3.1 to +5.2 ppb). The central value is slightly less than that in our original submission (+4.9 ppb), due to the exclusive use of sensitivity values from Fry et al. (2012) rather than also using hemispheric-scale sensitivity results from earlier studies. The impact from COVID lockdown CO emission reductions on methane is -1.3 ppb (range -0.8 to -1.8 ppb) (Table 2) and for NMVOC it is -0.7 ppb (-0.2 to -1.3 ppb). Inclusion of CO and NMVOC emission reductions counteracts about half of the estimated effect from NOx, so these are significant, as suggested by the reviewer. However, NOx remains the largest, and overall dominant, component.

For both NOx and NMVOC, accounting for spatial variations (using the four HTAP regions) in surface emissions increases the overall sensitivity of methane to NOx by +15% compared to using a simple global sensitivity. Spatial variations for CO make little difference, due to its longer lifetime.

| CO emission region | $\Delta CH_4/\Delta E_{CO}$ ppb($CH_4$)/Tg(CO) yr$^{-1}$ | 2020-2019 $\Delta E_{CO}$ Tg(CO) yr$^{-1}$ | $\Delta CH_4$ ppb($CH_4$) |
|---|---|---|---|
| Global | 0.0175 | -73.38 | -1.284 |
| Europe | 0.0167 | -6.09 | -0.102 |
| N. America | 0.0187 | -7.49 | -0.140 |
| E. Asia | 0.0178 | -24.58 | -0.438 |
| S. Asia | 0.0163 | -16.76 | -0.273 |
| Global - 4 regions | (0.0175) | -18.48 | -0.323 |
| Sum (+/-1 SD) | | | -1.3 (-0.8 to -1.8) |

**Table 1** Sensitivities of global methane mole fraction to changes in surface anthropogenic CO emissions ($\Delta CH_4/\Delta E_{CO}$, ppb($CH_4$)/Tg(CO) yr$^{-1}$) (Fry et al., 2012); estimated reductions in CO emissions in 2020 relative to 2019 (Tg(CO) yr$^{-1}$) (Lamboll et al., 2021); and resultant estimated changes in global methane mole fraction (ppb). We show global values, and values for the four HTAP regions. We use the global sensitivity value to compute a methane change resulting from the emissions changes outside the four regions. We sum methane changes from the four regions plus the rest of the world to yield a global value that accounts for the regional variations in sensitivity. We include an estimated uncertainty range based on the one standard deviation range in sensitivities found by the HTAP models (Fry et al., 2012).

| NMVOC emission region | $\Delta CH_4/\Delta E_{NMVOC}$ ppb($CH_4$)/Tg(C) yr$^{-1}$ | 2020-2019 $\Delta E_{NMVOC}$ Tg(C) yr$^{-1}$ | $\Delta CH_4$ ppb($CH_4$) |
|---|---|---|---|
| Global | 0.0407 | -15.65 | -0.636 |
| Europe | 0.0398 | -1.71 | -0.0681 |
| N. America | 0.0353 | -1.56 | -0.0551 |
| E. Asia | 0.0333 | -2.41 | -0.0803 |
| S. Asia | 0.0691 | -4.34 | -0.2999 |
| Global - 4 regions | (0.0407) | -5.63 | -0.2291 |
| Sum (+/-1SD) | | | -0.7 (-0.2 to -1.3) |

**Table 2** Sensitivities of global methane mole fraction to changes in surface anthropogenic NMVOC emissions ($\Delta CH_4/\Delta E_{NMVOC}$, ppb($CH_4$)/Tg(C) yr$^{-1}$) (Fry et al., 2012); estimated reductions in NMVOC emissions in 2020 relative to 2019 (Tg(C) yr$^{-1}$) (Lamboll et al., 2021); and resultant estimated changes in global methane mole fraction (ppb). We show global values, and values for the four HTAP regions. We use the global sensitivity value to compute a methane change resulting from the emissions changes outside the four regions. We sum methane changes from the four regions plus the rest of the world to yield a global value that accounts for the regional variations in sensitivity. We include an estimated uncertainty range based on the one standard deviation range in sensitivities found by the HTAP models (Fry et al., 2012).

| NOx emission region | $\Delta CH_4 / \Delta E_{NOx}$ ppb($CH_4$)/Tg($NO_2$) yr$^{-1}$ | 2020-2019 $\Delta E_{NOx}$ Tg($NO_2$) yr$^{-1}$ | $\Delta CH_4$ ppb($CH_4$) |
|---|---|---|---|
| Global | -0.1570 | -19.381 | 3.043 |
| Europe | -0.1001 | -2.650 | 0.265 |
| N. America | -0.1674 | -2.548 | 0.427 |
| E. Asia | -0.1560 | -4.399 | 0.686 |
| S. Asia | -0.3091 | -3.782 | 1.169 |
| Global - 4 regions | (-0.1570) | -6.002 | 0.942 |
| Sum (+/-1SD) | | | 3.5 (2.4 to 4.5) |

**Table 3** Sensitivities of global methane mole fraction to changes in surface anthropogenic NOx emissions ($\Delta CH_4 / \Delta E_{NOx}$, ppb($CH_4$)/Tg($NO_2$) yr$^{-1}$) (Fry et al., 2012); estimated reductions in NOx emissions in 2020 relative to 2019 (Tg($NO_2$) yr$^{-1}$) (Lamboll et al., 2021); and resultant estimated changes in global methane mole fraction (ppb). We show global values, and values for the four HTAP regions. We use the global sensitivity value to compute a methane change resulting from the emissions changes outside the four regions. We sum methane changes from the four regions plus the rest of the world to yield a global value that accounts for the regional variations in sensitivity. We include an estimated uncertainty range based on the one standard deviation range in sensitivities found by the HTAP models (Fry et al., 2012).

*Second, the calculation of the authors relies on the sensitivity of CH4 mixing to NOx emissions, taken from previous studies. Most of the studies cited are from the 2000s. The "baseline" emissions of NOx as well as other chemicals may have changed a lot from the early 2000s to 2020. I wonder if the sensitivity of global OH and CH4 to NOx emissions will also change, and if so change by how much, with the "baseline" emissions. This chemical system is known to be nonlinear.*

**Response**

This is also a good point. We fully appreciate that atmospheric OH chemistry is non-linear and the derived model sensitivities are somewhat dependent on the baseline emissions in the models, and that the baseline used in the HTAP study (and the earlier studies we used) differed from the 2020 emissions. However, we can only work with what we have available. As far as we are aware, there has been no published analysis of OH changes and impacts on methane from the second phase of the HTAP study, nor have there been more up-to-date regional analyses within other model inter-comparisons. Derwent et al. (2021) show that although chemical mechanisms used by global models have been developed and updated since 2000, they have not changed significantly. So we feel that the Fry et al. (2012) study is the most useful available study for both providing regional sensitivities, and also providing a model spread of sensitivities. We also feel that even though the system is non-linear, the sensitivities are unlikely to differ markedly from those derived from the HTAP study. Conducting a whole new HTAP style set of integrations with either a single model, or ideally a number of models, in order to analyse methane responses during the COVID lockdowns would be ideal, but that is beyond the scope of what we currently have available (see comments about HTAP phase 2 results below). In the revised version of the paper, we will only use the earlier studies to introduce the methodology; we will use sensitivity values for surface emissions sensitivities from Fry et al. (2012) as we consider this study to be the most up-to-date source of useful information.

*Third, independent observation evidence on reduced global OH, if there is, can be really powerful. The 5 ppb additional increase in methane mixing ratio translates roughly to a 3% decrease in global OH concentration if they were attributed entirely to reduced NOx emissions. This magnitude of decrease in global OH can have detectable signals on burden or distribution on many species besides CH4 , such as CH3CCl3. If these analyses are consistent with the author's hypothesis of NOx chemical feedback, it can really increase the confidence, though it may be a lot of work.*

**Response**

Again, we agree, and independent observational evidence of changes in OH would be very useful. However, after a quick look at recent data, including CH3CCl3, it is not immediately obvious that there are measurements available that can provide clear evidence of OH changes. So we think this is beyond our scope with this paper.

**Referee #2**

*1. My main concern is that the authors directly apply the sensitivity of CH4 to NOx emission changes estimated by previous studies for broad regions using different models for different periods. Since the OH chemistry is higher nonlinear, such estimation can lead to a large bias. For example, if we use the sensitivity of OH to NOx emission changes in the N. Hemisphere (-0.39) and the 16.72Tg NOx emission changes to estimate CH4 changes, we get 6.5Tg CH4 changes in the N. Hemisphere, much smaller than the 8.5Tg when considering sensitivity in 4 different regions. The sensitivity given by Wild et al. (2011) and Derwent et al. (2011) are estimated by perturbing the emission for the whole year, but the lock-down time-period are different in each country. For example, the emission reductions in China (East Asia) mainly occur during February, and gradually back to normal from April (Fig.6 in Miyazaki et al. (2021)). In winter the sensitivity of CH4 to OH may be much lower than in other seasons (low OH production and CH4+OH reaction rate). Thus apply the sensitivity estimated for the whole year may lead to overestimate of CH4 changes. In addition, most of the sensitivities in table 1 are estimated based on simulation for 2000 or earlier. Changes in global emissions from 2000 may influence the sensitivity of OH to precursor gases.*

*I agree that the reduction in NOx can contribute to the rising CH4 during 2020, but I don't think the 4.9ppb increase in the CH4 mixing ratio estimated in this study is reliable considering the nonlinearity in OH chemistry. Besides, emissions of other chemical species such as CO also changed during the lockdown period. So, the conclusion that "the NOx changes can account for all or most of the observed methane changes" cannot be supported by the simple calculation present in the manuscript. The changes in emissions are already available (Lamboll et al. 2021). I recommend the authors quantify the sensitivity of OH to emission reductions by conducting model simulations for 2020.*

**Response**

We agree that our method is a simplification, but we think that it is probably close enough to the right answer to be useful. Of course, the spatial and temporal structure of the emissions changes will influence the resulting perturbation to OH and methane. We hope that by including inter-model uncertainties from Fry et al. (2012) in our estimates of the regional and global sensitivities (see Tables 1-3) that we now convey a better sense of the level of uncertainty in our simple modelling approach. We agree that the statement that NOx changes can account for "all or most" of the observed methane changes was a bit too strong, and we will tone down our language in the revised version. We do feel that our simple modelling shows that changes in emissions (and in particular NOx) associated with the lockdowns can account for a significant component of the observed methane change. Future, more detailed modelling will be needed to better spatially and temporally resolve the contributions to the observed methane changes from emissions changes more precisely.

NB Wild et al. (2001) perturbed emissions for a whole year, but Derwent et al. (2001) perturbed emissions for a single month (January). No study has investigated seasonal variations in the methane response for surface emissions, but Stevenson et al. (2004) calculated impacts from aircraft NOx for January, April, July and October, and found a modest seasonality in response. The largest response was found in July, when the impact of aircraft NOx on methane was 10% larger than the annual mean value (Figure 2d and Table 4 of Stevenson et al., 2004).

*2. Most of the sensitivity of CH4 to NOx emissions changes listed in Table1 is not the original data that we can find from the references. I think the authors should clarify how they convert the data from the references to which is listed in table 1 in the supplementary.*

**Response**

We attempted to succinctly describe our methods for deriving the sensitivities in the original submission. We will add more details in the revised paper to clarify exactly where the values come from.

**Referee #3**

*While the findings are definitely interesting and make use of pre-existing model studies, for a research paper in Atmos. Chem. Phys., the analysis is not substantial enough. For example, the paper makes use of methane-to-NOx sensitivities based on very broad regions in the case of surface emissions (N. America, Europe, S. and E. Asia, Southern Hemisphere) and a global sensitivity scaling factor in the case of aircraft emissions, despite the spatial heterogeneity in sensitivity, as the authors themselves noted. In particular, the southern hemisphere is treated as a single region, despite the potential for increased sensitivity in low-NOx regions. More regional-scale sensitivities could be available through Phase 2 of the Hemispheric Transport of Air Pollution initiative, but the authors did not explore their applicability to support their hypothesis.*

**Response**

We agree, to some extent, with all these comments. The paper was originally submitted as an ACP Letter ([https://www.atmospheric-chemistry-and-physics.net/about/manuscript_types/acp_letters.html](https://www.atmospheric-chemistry-and-physics.net/about/manuscript_types/acp_letters.html)) which partly explains its brevity. We wished to alert the community to our findings rapidly, as we felt that they were of considerable interest – partly to stimulate others to do the more substantial research required to confirm (or refute) our suggestion. We agree that there may be important spatial structure in the sensitivities that will not be represented by using the broad HTAP regions. However, we think it is likely that the overall result will not be substantially different even if more detailed modelling is performed. Exploitation of the Phase 2 HTAP results is an excellent idea, but no analysis has yet been conducted on the responses of OH or methane within this study, so we would need to start from scratch. We are looking into this, but it is beyond the scope of this initial study. As explained above, we focussed on results from the first phase of HTAP presented by Fry et al. (2012) as we feel these are the best available results we can easily use.

One subtlety about the spatial variations is worth clarifying. The sensitivities we use, although representative of broad regions, do take into account the spatial distribution of emissions within those regions. The advantage gained by using smaller regions is that the folding of emissions change and sensitivity is performed at a higher resolution, so a more realistic answer is produced. The real sensitivities for a region will only differ strongly if the spatial distribution of emissions changes between the time of the experiment we use to diagnose the sensitivity and those during the 2020 lockdown. As argued above, whilst there have been changes over the last two decades, we don't think the sensitivities will be very different from those found by Fry et al. (2012). The emissions changes during lockdown also had spatial structures quite similar to the baseline emissions distributions, which also means the sensitivities used are appropriate.

*The second major consideration is that the potential impact of changes in commensurate emissions of carbon monoxide and/or volatile organic compounds during national lockdowns on the atmospheric methane growth rate was not quantified. Although NOx is clearly a contributing factor, it is not the sole influence and these other potential contributors to changes in methane growth rate were not assessed.*

**Response**

See our above response on this topic to Referee #1. We stressed in the original submission that NOx was not the sole influence, but one of several factors. We also stressed that we thought it was a major factor, and we maintain that assertion.

*Finally, the authors cite the paper of Lamboli et al. who estimate the emissions reductions due to COVID lockdowns and outline a protocol for global climate and Earth System Model simulations. First results on the climate response from these model simulations have been published and interactive chemistry was included in a number of models. Therefore, there is data available that could address the spatial heterogeneity of the methane sensitivities and the role of other emission changes that would add substantially to this analysis.*

**Response**

We did refer to some results from detailed simulations with interactive chemistry of the COVID lockdowns in our original submission (e.g., Weber et al., 2020; Miyazaki et al., 2021). We will include any more recent papers with relevant results in our revised version. However, we don't know of any current data that would address the impacts from different emissions components or regions on OH during lockdown, so we believe the HTAP results from Fry et al. (2012) to be the most relevant for estimating the component impacts on methane.

**Acknowledgements**

DS would like to thank PhD student Jize Jiang for technical assistance with some of the analysis.

**Reference** (in addition to those in the original submission)

Richard G. Derwent, David D. Parrish, Alex T. Archibald, Makoto Deushi, Susanne E. Bauer, Kostas Tsigaridis, Drew Shindell, Larry W. Horowitz, M. Anwar H. Khan, Dudley E. Shallcross (2021) Intercomparison of the representations of the atmospheric chemistry of pre-industrial methane and ozone in Earth system and other global chemistry-transport models, Atmospheric Environment, Volume 248, 118248, https://doi.org/10.1016/j.atmosenv.2021.118248

---

## Author Response (AR2)

**Response to reviewers**

Reviewer's comments are shown with a *grey background and in italics, like this*.

**Reviewer #1**

*The manuscript has been greatly improved by accounting for the effect of reduced CO and NMVOC emissions and by including an ensemble of simulations to characterize uncertainties. I have a few minor suggestions:*

*Line 97-98: Minor sinks of methane include oxidation by Cl. How is that accounted for in the methane lifetime calculation? Probably not important for the results, but better to include for completeness.*

Many thanks for pointing out this omission. We have now included the Cl sink in the whole atmosphere methane lifetime calculations. It shortens the lifetime and perturbation lifetime (as expected), but has an only minor impact on the results (also as expected).

*Fig. 1 Caption: better to briefly describe what perturbation is performed in the caption (e.g., 20% reduction of global anthropogenic emissions including NOx, CO, and NMVOC), so a reader does not have to look for the information in the text.*

We have added clarification to the captions for Figure 1 and Figure 2.

*Line 133: Remove "As for NOx,"*

We have removed.

**Reviewer #2**

*In the revised manuscript, the authors include the influences from CO and NMVOCs. But the results are still estimated using the sensitivity for board [sic: broad] regions over 2000. For different sectors, the emission reductions during the COVID-19 lockdown are different. Thus, the emission reductions may be heterogeneously distributed. I agree with the authors that the emission changes can influence OH and methane chemical sink. However, considering the nonlinear OH chemistry, and the spatial and temporal heterogeneous emission changes, I still think such simplifications are not enough for estimating the OH and CH4 changes in response to emission changes. The main conclusion may be right, but I suggest the model simulations using the emission inventory given by Lamboll et al. (2021) are needed to support the conclusion.*

We appreciate this reviewer's concerns and agree with many of these points. We also agree that new model simulations using the Lamboll et al. (2021) inventories would be very useful to support (or not support) our findings. However, this, in our opinion, would be a different study altogether, and is for the future. We acknowledge that OH chemistry is non-linear, and that by using HTAP simulations with a base year of 2001, that this somewhat compromises our results, since emissions have evolved since 2001. We also acknowledge that the emissions changes seen during the lockdowns will differ spatially from the emissions changes used in the HTAP experiments. On the more positive side, we note that the -20% emissions changes applied in the HTAP experiments are remarkably similar in magnitude to the emissions reductions estimated over the lockdowns, so that at least this source of error should be minimised. The HTAP study is also incredibly useful in providing a multi-model ensemble, allowing us to assess differences in sensitivities across several models, and estimate an uncertainty range. In addition, the regional experiments performed within

HTAP are unique for a co-ordinated global multi-model study, and this provides further incredibly useful extra information, allowing us to break down regionally the origin of the lockdown influence on methane. To perform a multi-model HTAP-like regional set of experiments with the Lamboll et al. (2021) emissions would be ideal to answer these questions in more detail, however that would be a major effort, and is something the community should consider for the future (perhaps this paper could provide a basis for such a study). Until such a study has been done, it is impossible to say whether the simplified estimations presented here are more or less realistic. We argue that the results we present here are useful, but have several caveats, as pointed out by the referee, and as we have clearly acknowledged in our paper. One of the points of writing the paper (as with most or all papers?) was to stimulate the community to perform further research to confirm or refute our results, as they appear to be important (we note both referees indicate the scientific significance of our results as outstanding).

*Besides, in my last comments, I mentioned that the author should clarify how they estimate the results from the original data given by Fry et al. (2021). I cannot find it in the revised manuscript. Also, I suggest the authors list the sensitivity of CH4 changes to NOx/CO/NMVOC emissions used to estimate the values in table 2 in the manuscript.*

We have now added some Supplementary Information that hopefully comprehensively clarifies how we have calculated our results using the original data in Fry et al. (2012). The supplementary information is a spreadsheet with multiple pages that steps through the calculations from the raw model data used in HTAP to produce Figure 4 in Fry et al. (2012). We recreate that figure, which shows the equilibrium response of methane to emissions changes in the HTAP simulations, then extend the analysis to produce the sensitivities in Figures 2, 3 and 4 of our paper. The values for the plotted sensitivities in these figures are given in the supplementary Table S10. We appreciate that our methodology, whilst fundamentally relatively simple, has multiple non-obvious steps, and we hope this supplementary information provides sufficient clarification to make it possible for anyone interested in reproducing or analysing our results or developing our methods to do so.

**Note to the editor and reviewers**

In revising the analysis to satisfy the reviewers, we found some (ultimately relatively minor) additional errors in our analysis contained in the revised submission of 3rd August. We have corrected those errors in this version, but it has not significantly altered our overall results.

The first error was that we realised the HTAP NOx experiments were 20% perturbations to *all* anthropogenic emissions, not just surface emissions. This meant aviation was already included, and by adding it we were accounting for it twice. In fact, this was a bit more complicated, as the surface emissions reductions during lockdown were about 15%, whereas the aviation reductions were 23%, so about 8% of the aviation reduction did still need to be added. We have attempted to clearly describe how we have done this in the revised version.

The second error was that we realised we were using the incorrect HTAP emissions changes to produce the sensitivities in Figures 2-4. The supplementary information in Fiore et al. (2009) provides both total emissions data and anthropogenic emissions data. We had been erroneously using the total emissions rather than the anthropogenic emissions to calculate the 20% emissions changes applied in HTAP. The sensitivities calculated now are more like the estimates in the initial response to reviewers presented in the ACPD discussion.

---

## Author Response (AR3)

**Response to review by editor**

Many thanks for your review:

*Thank you for the newly revised manuscript and your response to the referees' comments. The comments by Reviewer #1 have been well answered, but I am not satisfied with your response to Reviewer #2's concerns about how the nonlinear response of atmospheric OH and CH4 to spatially and temporally heterogeneous emissions might affect your results and conclusions. I understand that it would be a major effort to follow the referee's suggestion and include the Lamboll et al. (2021) emissions scenario, something that you are planning for the future. In your response to the referee, you acknowledge that the non-linear behaviour of OH could alter your conclusions if spatially and temporally heterogeneous emissions are taken into account. I think this point must be explicitly stated in your paper. Given these and the other caveats that you already mention in the manuscript, I think the wording in the title of the paper "COVID-19 lockdown emission reductions can explain over half of the coincidental increase in global atmospheric methane" is too strong. I suggest toning down the statement in the title, for example, saying "emission reductions have the potential to explain ..." instead of " emission reductions can explain...". Provided that the conclusions and title are toned down as mentioned above, I accept the manuscript for publication in ACP.*

We have adjusted the title and conclusions as you suggest, and hope the paper is now acceptable for publication.